# Relative income within the household, gender norms, and well-being

Rania Gihleb[1☉], Osea Giuntella [1☉]*, Luca Stella[2☉]

**1** Department of Economics, University of Pittsburgh, Pittsburgh, PA, United States of America, **2** Department of Economics, Catholic University of Milan, Milan, Italy

☉ These authors contributed equally to this work.
* osea.giuntella@pitt.edu

## Abstract

This study examines the effects of relative household income on individual well-being, mental health, and physical health in Germany. Consistent with previous studies, we document a dip in the distribution of households in which the wife out-earns the husband. Using a regression discontinuity design, we show that husbands in couples in which the wife earns just more exhibit lower satisfaction with life, work, and health, and report worse physical health. Women in these couples report lower satisfaction with life and health, and worse mental health. Results on life, work, and health satisfaction among women are more pronounced in West Germany, consistent with previous evidence of gender norm differences between East and West Germany.

**Data Availability Statement:** Data is available from the OSF page https://osf.io/k3bp7/ with references to the data sources and the version of the SOEP data used in the analysis. The zipped folder contains the data and code used in the analysis.

## Introduction

Over the last few decades, the share of couples in which women earned more than their male partners has increased significantly. In 2000, only 13% of married women in the US earned more than their husbands; in 2017, this rate was above 31% [1]. This trend is also observed in other countries [2] and mirrors the reversal of the gender gap in education [3, 4].

These patterns challenge the traditional norms of male breadwinners and may affect couples' well-being and mental health [1]. Although many studies have analyzed the effects of relative income on labor market outcomes and marital behavior [5–7], the relationship between relative income and the well-being and mental health of husbands and wives has received less attention [8, 9]. This study uses longitudinal survey data from Germany to examine how spousal relative income affects the individual well-being and health.

Germany provides a unique research setting because of its history and the prevalence of different gender norms in East and West Germany [7, 10–12]. In the German Democratic Republic (GDR), female labor force participation was considered an essential element of socialist production. The provision of public childcare led to an incredible increase in female labor force participation, peaking at 89% in 1989 [13]. These trends have gradually eroded the male breadwinner norm [14]. At the same time, for a long time, the Federal Republic of Germany (FRG) enacted policies that implicitly supported traditional marriage and a single-earner family model [14].

**Funding:** The author(s) received no specific funding for this work.

Given this context, it is unsurprising that in East Germany, the share of women earning more or about the same as their male partners was historically larger. In 1992, in East Germany, almost 30% of women earned more than or approximately the same amount as their male partners, compared to 16% in West Germany. The gender wage gap between West and East Germany has declined substantially. In West Germany, the share of households in which the wife out-earns the husband increased from 12% in 1984 to 23% in 2020. Nonetheless, considerable differences persist; as of 2020, in East Germany, 32% of women earn as much as their male partners or more.

Most previous studies examined the relationship between gender identity and labor market outcomes. In this study, we instead focus on the impact of relative income within the household on individuals' mental well-being and physical health and explore how the different gender norms in East and West Germany may characterize this relationship.

Our work relates to a growing number of studies analyzing the importance of relative spousal income and the role of gender norms. In their study using US data, Bertrand et al. (2015) [5] analyzed the impact of relative income on labor market outcomes, home production, marital satisfaction, and divorce. They showed a marked decline in the distribution of households to the right of the 50% threshold in the income share distribution of wives within households, that is, when they out-earn their husbands. The authors provided evidence consistent with the view that gender identity norms significantly affect various social and economic outcomes. Couples in which the wife earns more than the husband are less satisfied with their marriage and are more likely to divorce. Furthermore, using time-use data, they find that the gender gap in non-market work is larger when wives earn more than their husbands. More recently, other studies that attempt to replicate the findings of Bertrand et al. (2015) have yielded mixed results. No evidence of discontinuity has been detected in Sweden or Finland [15, 16]. Binder et al. (2022) [17] find that simple models of assortative matching can replicate the observed distribution of spousal earnings differences, in which very few wives out-earn their husbands. Their results question the role of gender norms in explaining the distribution of spousal earnings in the US. In contrast, Sprengholz et al. (2022) [7] analyzing SOEP data provide evidence consistent with the male breadwinner norm in West (but not East) Germany.

In particular, we add to the few studies investigating the role of higher female earnings and the impact of relative income on psychological distress and mental health [1, 18–20]. Two studies are particularly similar to ours: Getik (2022) [8] and Salland (2018) [9]. Getik (2022) uses administrative data from Sweden and finds that mental health is positively associated with own and spousal income but negatively linked to the wife's relative income, with an 8–12% increase in mental health diagnosis when comparing couples in which the wife earns more than the husband. Salland (2018) [9] used SOEP data to study the effect of within-household comparison on individual life satisfaction. He finds that a primary breadwinner wife decreases a couple's individual happiness by 8%.

Our contribution to these studies is twofold. First, Germany provides an interesting context for studying the relationship between gender norms and well-being. Campa and Serafinelli (2019) [21] and Boelmann et al. (2021) [22] document the substantial heterogeneity in gender role attitudes and female labor supply between East and West Germany. Furthermore, Lipmann et al. (2020) [23] highlight the role of institutions in undoing the male breadwinner norm in East Germany. Previous studies have found evidence that gender identity influences the labor supply of full-time working women, but only in West Germany [7]. The SOEP data allow us to explore the different roles of relative household income on well-being and health in cultures characterized by different gender norms. This aspect has not been explored in previous studies.

Second, while some previous work using data from Scandinavian countries has the clear advantage of large and granular administrative datasets, SOEP data enable us to examine a wide range of metrics of well-being and health. Salland (2018) [9] also used SOEP data, but his work mostly focused on life satisfaction and did not examine the differences between East and West Germany. We explore the effects on satisfaction with life, work, and health, as well as other metrics of self-reported mental and physical health.

Our results shed light on the role that institutions and policies can play in shaping norms, significantly affecting labor market opportunities and individuals' well-being and health.

## Data and empirical specification

### Data

We draw our data from the German Socioeconomic Panel (SOEP), a longitudinal dataset of the German population containing information on a rich set of household and individual socioeconomic characteristics since 1984 [24].

The SOEP contains several metrics for self-reported satisfaction with various domains. In our main analysis, we focus on life satisfaction, satisfaction with work, and health satisfaction. Individual satisfaction is measured on an 11-point Likert scale ranging from 0 (very dissatisfied) to 10 (very satisfied). The SOEP also includes several measures of individual health. In our main analysis, we consider two metrics of physical and mental health as outcome variables. The 12-Item Short Form Health Survey (SF-12) is used to measure physical and mental health. Since 2002, the SF-12 questionnaire has been administered every two years. It includes 12 Likert-scale questions on various health aspects grouped into two broad summary scales for physical and mental health [25]. These summary scales take on continuous values between 0 and 100, with a mean of 50 and a standard deviation of 10. Higher scores indicate better health. In the analysis, for ease of interpretation, we standardize these metrics to have a mean of zero and standard deviation of one.

In secondary analysis in the S1 Appendix, we explore individuals' satisfaction with different life domains (i.e., family, housework, sleep and childcare), and we use the standard self-assessed health (SAH). For the SAH, respondents are asked to rate their current health status on a 5-point scale, ranging from "bad" and "poor" to "satisfactory," "good," and "very good." We use this information to create a dummy for poor health, equal to one if the subjective health status is less than good and zero otherwise.

To construct the running variable, we use information on annual earnings including wages and salaries from main job for husbands and wives. We restrict the analysis to married dual-earner couples aged 18–64 at the time of the survey. Furthermore, we constrain our sample to couples for whom information on all observables are not missing. Summary statistics are reported in the Table A.1 in S1 Appendix.

### Empirical specification

We use a regression discontinuity (RD) design [26, 27] to identify the effects of relative spousal income on the well-being and health of respondents. Formally, we estimate the following equation:

$$Y_{iht} = \beta_1 I[E_{iht}^{Wife} \geq E_{iht}^{Husband}] + \beta_2 (E_{iht}^{Wife} - E_{iht}^{Husband})$$
$$+ \beta_3 (E_{iht}^{Wife} - E_{iht}^{Husband}) \times I[E_{iht}^{Wife} \geq E_{iht}^{Husband}] + \eta_i + \epsilon_{iht} \quad (1)$$

where the index $iht$ denotes individual $i$ in couple $h$ interviewed in year $t$. $Y_{iht}$ is a metric of individual well-being and health. $I[E_{iht}^{Wife} \geq E_{iht}^{Husband}]$ is an indicator variable for individuals

living in households in which the wife earns more than the husband, and the difference $(E_{iht}^{Wife} - E_{iht}^{Husband})$ is our running variable defined in intervals of €1,000. The coefficient of interest is $\beta_1$, which captures the effect of being in a couple where the wife just out-earns the husband. $\eta_i$ are the individual fixed effects, which absorb the influence of any time-constant individual heterogeneity. Finally, $\epsilon_{iht}$ represents an idiosyncratic error term. The standard errors are clustered by relative income bins (in €1,000 intervals). Adjusting for clustering at the individual level does not alter the significance of our results. For each outcome, we used the optimal bandwidth using the mean square error (MSE) selection criteria [28]. In the S1 Appendix, we include in Eq (1) a vector of controls such as individuals' age, education, and earnings of both partners, as well as state- and year- fixed effects.

A typical concern of any RD analysis is that couples on different sides of the threshold may be selected concerning characteristics that correlate with our outcomes of interest. Specifically, we may be concerned that individuals sort based on health characteristics or the determinants of well-being and health outcomes. To partially mitigate this concern, we conduct balancing checks on our covariates as well as outcomes. For this analysis, we used the covariates measured in the first year of marriage (see also Getik 2022 [8]), and outcomes measured in the first year available in the SOEP before marriage and restricted the sample to individuals under 45. We present the results of this analysis in Tables A.2 and A.3 in S1 Appendix. Overall, we find that baseline covariates and baseline outcomes are balanced both when constructing the running variable using the first year of marriage (see Panel A of Table A.2 and column 1 of Table A.3 in S1 Appendix), or when considering all the available years in the data (see Panel B of Table A.2 and column 2 of Table A.3 in S1 Appendix).

## Results

As in Bertrand et al. (2015) [5], we find a dip in the distribution of households where the wife just out-earns the husband (see Fig 1).

We then turn to analyze the relationship between relative income within the household and individual well-being as well as mental and physical health.

Figs 2 and 3 graphically document this relationship among men and women, respectively. In couples in which wives out-earns their husbands, life satisfaction among men is significantly lower. We also find that men are generally less satisfied with their work (see Fig A.1 in S1 Appendix). Women's overall life satisfaction appears lower when they earn barely more than their husbands (see Fig 3 and Fig A.2 in S1 Appendix).

Table 1 shows the magnitudes of the RD estimates among men and women. Overall, men report lower life satisfaction levels. The effect is relatively small (0.07 standard deviations) but economically sizeable (see column 1 of Panel A). In our sample, the point estimate would be equivalent to approximately one-third of the effect of having a college degree on life satisfaction or approximately 10% of the negative effect of unemployment. There is also evidence of a mild decline in satisfaction with work (see column 2). Among women, overall life satisfaction declined by 0.09 standard deviations (see column 1 of Panel B). When examining satisfaction with the different life domains among men, we find similar declines in satisfaction with family, satisfaction with housework and satisfaction with sleep (see Panel A of Table A.4 in S1 Appendix). Among women, we find evidence of an increase in satisfaction with childcare, but a parallel decrease in satisfaction with housework (see Panel B of Table A.4 in S1 Appendix). There is, instead, no evidence of effects on other domains of life.

Exploring the impact on mental and physical health, we find evidence of a decline in satisfaction with health, and physical health among men, while no evidence of any effect on mental health (see Fig 2 and A.1 in S1 Appendix, and Panel A of Table 1). Instead, among women

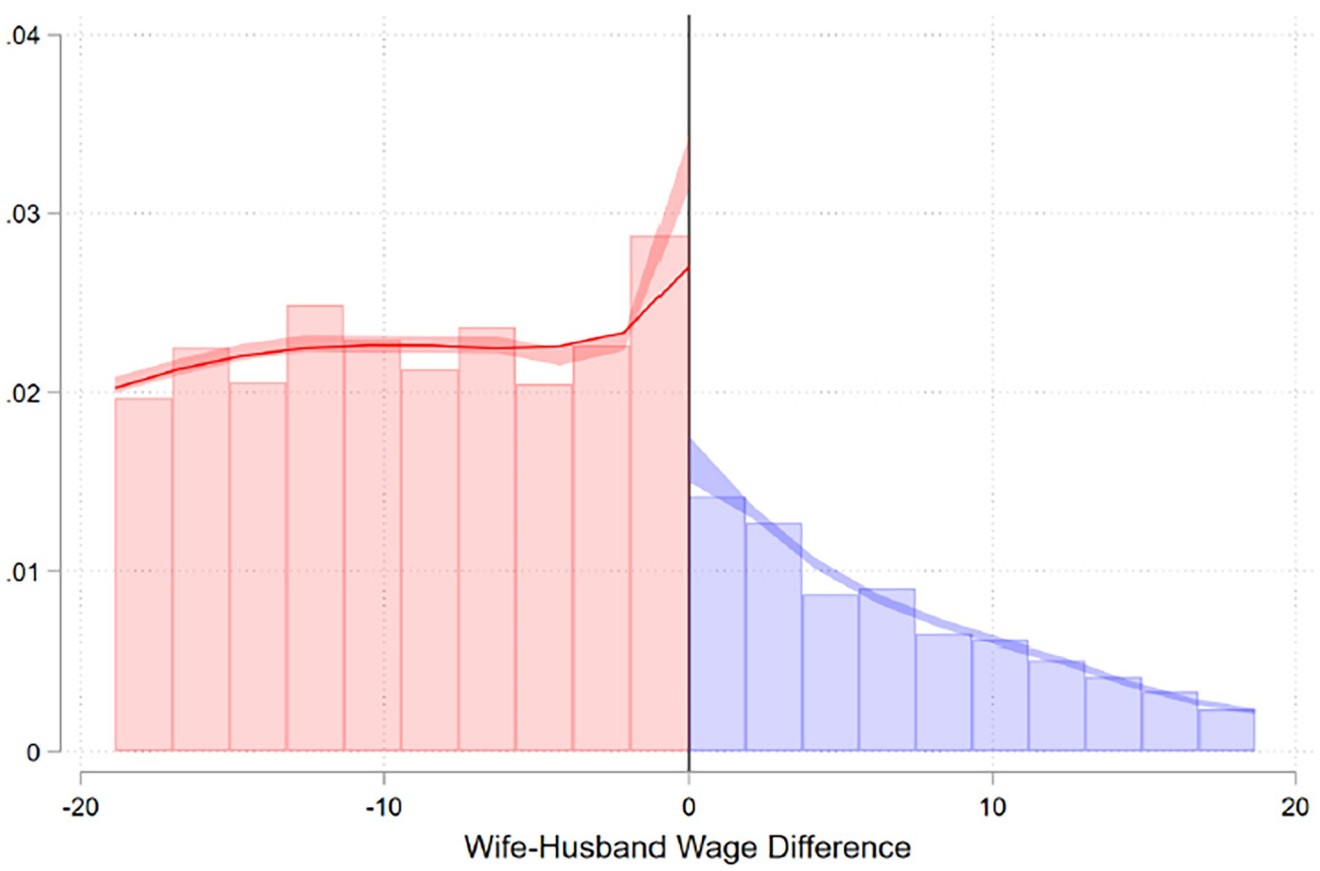

**Fig 1. Running variable density.** The figure shows the distribution of couples with respect to the gender difference in wage within the household (in €1,000).

there is an overall decline in both satisfaction with health and mental health (see Fig 3 and Fig A.2 in S1 Appendix, and Panel B of Table 1). Women exhibit a 0.08 standard deviation decline in satisfaction with health and a 0.05 standard deviation reduction in mental health.

In Table 2 we explore the heterogeneity of the effect of interest between East and West Germany. In this case, we add to Eq (1) an interaction term between a dummy variable that identifies individuals in couples where the wife out-earns the husband with an indicator variable for residing in West Germany. Furthermore, we control for a dummy variable for individuals residing in West Germany, and we allow the running variable to have different effects in East and West Germany. When considering the effect on life satisfaction for men, we find no evidence of significant differences between East and West Germany (see column 1 of Panel A). However, when considering women, we find that the negative effects on life satisfaction are significantly more pronounced (in absolute value) in West Germany (see column 1 of Panel B). Moreover, while for men the effects on satisfaction with work, satisfaction with health as well as mental and physical health are not significantly different between East and West Germany (see columns 2 to 5 of Panel A), for women in West Germany we find significant negative effects on satisfaction with work and satisfaction with health and no evidence of significant differences in mental and physical health (see columns 2 to 5 of Panel B). These results are partially consistent with previous studies documenting substantial cultural differences in gender norms and female labor supply between East and West Germany [21–23] and

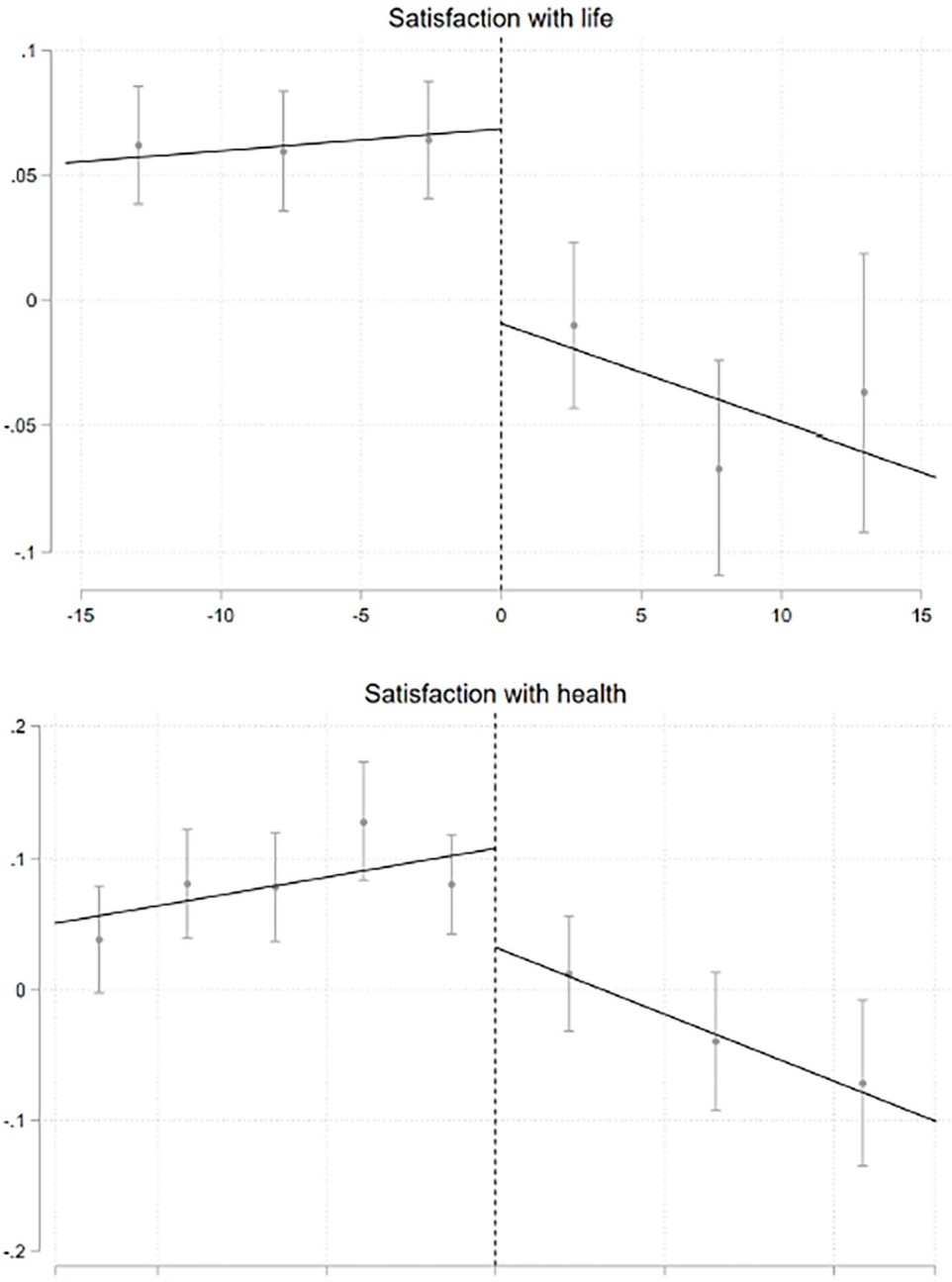

**Fig 2. Gerder norms, well-being and health, men.** This figure describes the relationships between satisfaction in different life domains, health (mental and physical) among men and the gender income differences within households. The standard errors are clustered by relative income bins (in €1,000 intervals). All specifications include individual fixed effects. For bandwidth and binning selection, we use the MSE-optimal bandwidth and the IMSE-optimal evenly-spaced method [26]. 95% confidence intervals are reported in bars.

with previous work analyzing how gender identity affects female labor supply among women in West Germany [7].

While our main analysis focused on life and work satisfaction, physical and mental health, in the Appendix, we report results on a broader set of outcomes exploring the different

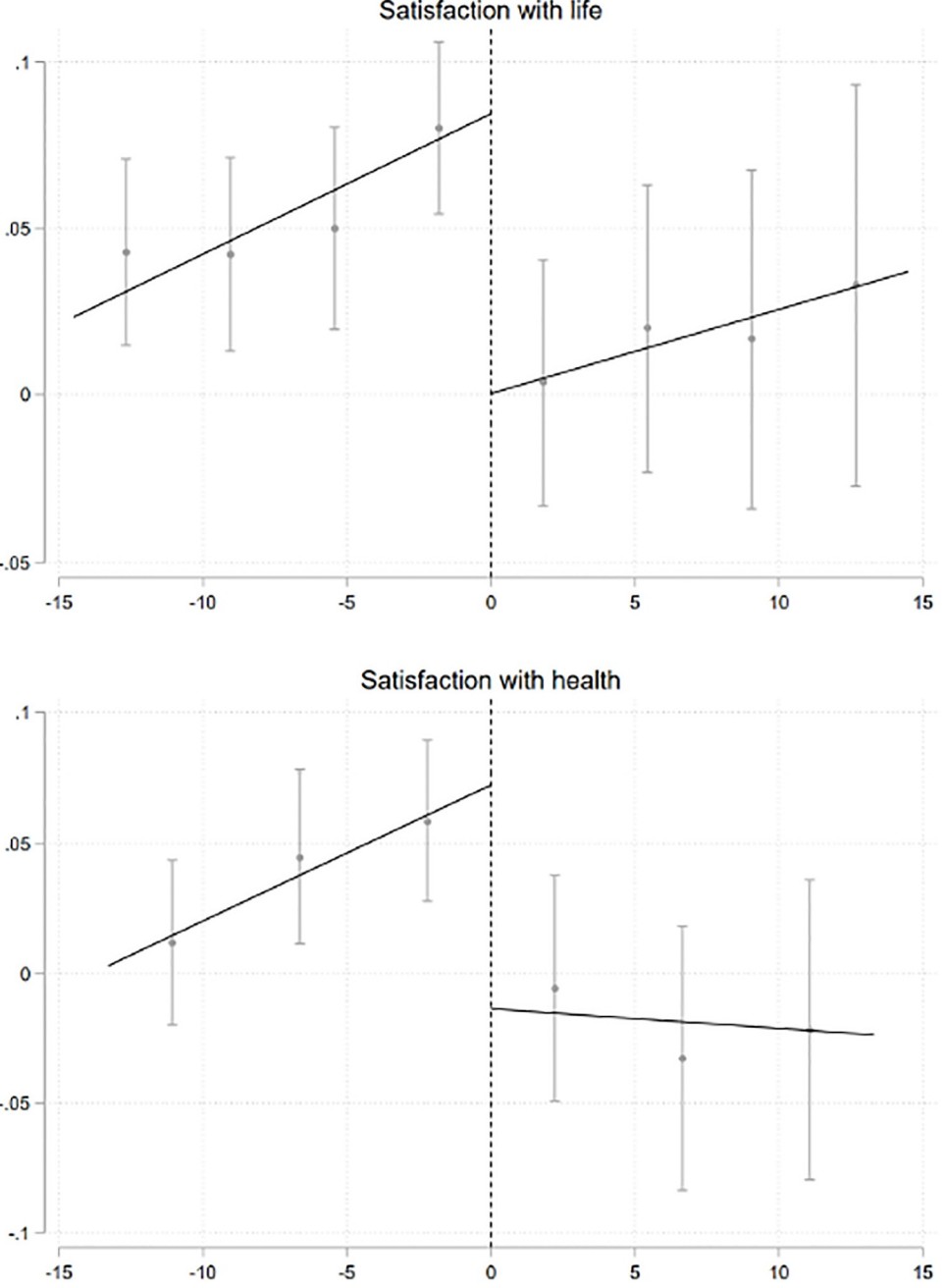

**Fig 3. Gerder norms, well-being and health, women.** This figure describes the relationships between satisfaction in different life domains, health (mental and physical) among women and the gender income differences within households. All specifications include individual fixed effects. The standard errors are clustered by relative income bins (in €1,000 intervals). For bandwidth and binning selection, we use the MSE-optimal bandwidth and the IMSE-optimal evenly-spaced method [26]. 95% confidence intervals are reported in bars.

**Table 1. RD estimates: Gender norms, well-being, and health—Including individual FE.**

| Dep. var.: | (1) | (2) | (3) | (4) | (5) |
|---|---|---|---|---|---|
| | **Satisfaction with life** | **Satisfaction with work** | **Satisfaction with health** | **Mental health** | **Physical health** |
| | | | Panel A: Males | | |
| RDD wife earns more | -0.074** | -0.066*** | -0.061* | 0.009 | -0.050*** |
| | (0.033) | (0.019) | (0.032) | (0.029) | (0.014) |
| Observations | 78,735 | 74,540 | 78,585 | 24,465 | 24,465 |
| Mean of dep. var. | 7.187 | 7.108 | 6.801 | 0 | 0 |
| Std. dev. of dep. var. | 1.725 | 2.041 | 2.152 | 1 | 1 |
| | | | Panel B: Females | | |
| RDD wife earns more | -0.093*** | -0.013 | -0.083*** | -0.050** | 0.005 |
| | (0.017) | (0.033) | (0.019) | (0.021) | (0.022) |
| Observations | 78,765 | 70,400 | 78,608 | 24,412 | 24,412 |
| Mean of dep. var. | 7.264 | 7.104 | 6.827 | 0 | 0 |
| Std. dev. of dep. var. | 1.732 | 2.061 | 2.151 | 1 | 1 |

*Notes* - Standard errors are reported in parentheses and are clustered by relative income bins (in €1,000 intervals). All specifications include individual fixed effects. For each outcome, we used the optimal bandwidth using the mean square error (MSE) selection criteria and robust inference following Calonico et al. (2014).

*Significant at 10 per cent;

** Significant at 5 per cent;

***Significant at 1 per cent.

**Table 2. RD estimates: Gender norms, well-being, and health—West and East Germany—Including individual FE.**

| Dep. var.: | (1) | (2) | (3) | (4) | (5) |
|---|---|---|---|---|---|
| | **Satisfaction with life** | **Satisfaction with work** | **Satisfaction with health** | **Mental health** | **Physical health** |
| | | | Panel A: Males | | |
| RDD wife earns more West | -0.005 | 0.051 | 0.030 | -0.025 | 0.008 |
| | (0.042) | (0.031) | (0.033) | (0.017) | (0.017) |
| Observations | 78,735 | 74,540 | 78,585 | 24,465 | 24,465 |
| Mean of dep. var. | 7.187 | 7.108 | 6.801 | 0 | 0 |
| Std. dev. of dep. var. | 1.725 | 2.041 | 2.152 | 1 | 1 |
| | | | Panel B: Females | | |
| RDD wife earns more West | -0.091*** | -0.043*** | -0.084*** | -0.010 | 0.021 |
| | (0.013) | (0.017) | (0.022) | (0.026) | (0.025) |
| Observations | 78,765 | 70,400 | 78,608 | 24,412 | 24,412 |
| Mean of dep. var. | 7.264 | 7.104 | 6.827 | 0 | 0 |
| Std. dev. of dep. var. | 1.732 | 2.061 | 2.151 | 1 | 1 |

*Notes* - Standard errors are reported in parentheses and are clustered by relative income bins (in €1,000 intervals). All specifications include a dummy for residing in West Germany, the earning-difference within the couple, a dummy indicating whether the woman out-earns the husband, and individual fixed effects. For each outcome, we used the optimal bandwidth using the mean square error (MSE) selection criteria and robust inference following Calonico et al. (2014).

*Significant at 10 per cent;

** Significant at 5 per cent;

***Significant at 1 per cent.

domains of life satisfaction (see Tables A.4 and A.5 in S1 Appendix). Furthermore, as previously mentioned, Tables A.6 and A.7 in S1 Appendix present estimates that control for a set of time-varying covariates. Results are similar to the baseline specification.

## Conclusion

In this study, we take advantage of longitudinal data from Germany and use a regression discontinuity design to study how relative spousal income affects individuals' well-being and health and how the prevalence of different gender norms in West and East Germany may mediate this relationship. In couples where the wife earns slightly more than the husband, husbands report worse satisfaction with life, work, and health. Women also report worse satisfaction with life and health, and lower levels of mental health when out-earning their husbands. We do not find any statistically significant differences between East and West Germany when considering mental and physical health. However, the results on life satisfaction, satisfaction with work and health are larger among women in West Germany. These findings partly align with earlier research that has identified significant differences in gender norms and the participation of women in the workforce between Eastern and Western Germany.

## Supporting information

**S1 Appendix.**
(PDF)

## Acknowledgments

We are thankful to the Editor and the Reviewers for their helpful comments. We benefited from the feedback in seminars at the University of Pittsburgh.

## Author Contributions

**Conceptualization:** Rania Gihleb, Osea Giuntella, Luca Stella.

**Formal analysis:** Luca Stella.

**Investigation:** Rania Gihleb, Luca Stella.

**Methodology:** Rania Gihleb, Osea Giuntella, Luca Stella.

**Writing – original draft:** Rania Gihleb, Osea Giuntella, Luca Stella.

**Writing – review & editing:** Rania Gihleb, Osea Giuntella, Luca Stella.

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
