## [Decision Letter · Decision Letter 0]

6 Mar 2024

PONE-D-24-02294Relative Income within the Household, Gender Norms, and Well-beingPLOS ONE

Dear Dr. Giuntella,

Thank you for submitting your manuscript to PLOS ONE. After careful consideration, we feel that it has merit but does not fully meet PLOS ONE’s publication criteria as it currently stands. Therefore, we invite you to submit a revised version of the manuscript that addresses the points raised during the review process.

We look forward to receiving your revised manuscript.

Kind regards,

José Alberto Molina

Academic Editor

PLOS ONE

2. Please update your submission to use the PLOS LaTeX template. The template and more information on our requirements for LaTeX submissions can be found at http://journals.plos.org/plosone/s/latex

4. We notice that your supplementary figures and tables are included in the manuscript file. Please remove them and upload them with the file type 'Supporting Information'. Please ensure that each Supporting Information file has a legend listed in the manuscript after the references list.

Reviewers' comments:

Reviewer's Responses to Questions

**Comments to the Author**

1. Is the manuscript technically sound, and do the data support the conclusions?

Reviewer #1: Partly

Reviewer #2: Partly

2. Has the statistical analysis been performed appropriately and rigorously? 

Reviewer #1: No

Reviewer #2: Yes

3. Have the authors made all data underlying the findings in their manuscript fully available?

Reviewer #1: Yes

Reviewer #2: Yes

4. Is the manuscript presented in an intelligible fashion and written in standard English?

Reviewer #1: Yes

Reviewer #2: Yes

5. Review Comments to the Author

Reviewer #1: The paper studies how relative household income affects individual wellbeing, physical health, and mental health in Germany using data from the German Socioeconomic Panel (SOEP). In particular, the authors use a Regression-Discontinuity approach to identify a causal effect. The authors find that, for men, earning less than their female partner has a negative effect on satisfaction with life, family, sleep, and health (both physical and mental). Furthermore, results on satisfaction with life and health seems to be mostly driven by respondents form West Germany, which the authors interpret as being the result of gender norm differences between East and West Germany.

I have some major concerns (even though some could easily be addressed with more clarifications) about the econometric analysis and the interpretation of the results.

1) The regression model reported in equation (1) only includes the dummy for being above the cut-off and the running variable. It does not include a constant term and an interaction between the dummy and the running variable. I think both are (and should be) typically included in a Regression-Discontinuity design unless specific assumptions are made (see for example Imbens and Lemieux 2008 or Calonico et al. 2017 - which the authors also refer to). It’s thus not clear whether the estimates reported in the paper are derived from equation (1), which I don’t think identifies the desired effect, or from using the “rdrobust” package from Calonico and co-authors, which would include the two terms and therefore there is a mistake in the reported estimated equation.

2) My second concern is related to the fact that people can endogenously sort on the two sides of the cut-off or that the estimates could reflect, for example, reverse causality.

a. The authors alleviate the concern by conducting a balancing check on both covariates (measured in the first year of marriage) and outcomes (as measured before marriage). However, it’s not entirely clear to me how this check is conducted. Given the panel structure of the data, the same person could be above or below the cut-off at different points in time. In which period is the running variable measured to decide whether a person is part of the ‘treated’ or ‘control’ group in the balancing check? Is the running variable also measured in the first year of marriage?

b. I think that the inclusion of individual fixed-effects also help alleviating this second concern, but it is currently left as a robustness exercise. I think it should be presented as the main analysis because (i) it accounts for the fact that unobserved heterogeneity might be correlated with sorting (ii) it exploits variation over time and within individual regarding the placement around the cut-off.

3) Moreover, point estimates and significance levels often change considerably when including fixed-effects, to the point that the interpretation of the results changes. For example, I find it difficult to reconcile the large, negative and significant effect on life satisfaction for men with the non-significant effect on work satisfaction – if the story is really about relative earned income. When individual fixed-effects are included, instead, the estimate is more precise and there is evidence of a negative and significant effect on work satisfaction, which seems more in line with the proposed story. On the other hand, the estimated effect on mental health for men changes considerably and it is zero when individual fixed-effects are included. Similarly, after including fixed-effects, there is little evidence of a negative effect on housework, work and sleep satisfaction for women.

4) The authors claim that “For both men and women, the effects on life satisfaction and mental well-being are mostly driven by respondents in West Germany”, but it seems that the results for men are not so conclusive.

a. The point estimates for life satisfaction for men in East and West Germany are virtually identical in Table A.1 (-0.079 vs. -0.082), even if only significant for West Germany (also, the East German sample is much smaller, which could explain why the estimate is less precise). It’s also surprising that the estimate for the entire sample (-0.139) it's much smaller than the ones for East and West Germany and does not lie between them. I cannot find the estimates by region when individual fixed-effects are included.

b. For mental health, the point estimate in Table A.2 for men it’s even larger in magnitude in East Germany (-0.061 vs. -0.037), and not significant in both cases.

5) In Section 2 the authors write “We report the estimates obtained using the triangular kernel to recover the average effects using the full bandwidth ….” However, in the notes to the tables, they write “For each outcome, we used the optimal bandwidth using the mean square error (MSE) selection criteria”. It’s thus not clear which bandwidth is used, or why we would want to use a non-optimal one.

References

Calonico, S., Cattaneo, M.D., Farrell, M.H., Titiunik, R., 2017. rdrobust: Software for regression-discontinuity designs. The Stata Journal 17, 372–404.

Imbens, G. W., Lemieux, T., 2008. Regression discontinuity designs: A guide to practice. Journal of Econometrics 142, 615-635.

Reviewer #2: Summary:

Utilizing data from the German Socio-Economic Panel, the authors examine how household relative earner roles affect individuals’ overall life satisfaction and satisfaction in various domains such as work, family, sleep, and health. The authors first report a discernible decline in the prevalence of couples where the wife assumes primary breadwinner responsibilities. Employing a regression discontinuity design methodology, the authors compare life satisfaction among individuals close to the threshold where the wife out-earns the husband. Their findings reveal that within couples where the wife out-earns the husband, men and women exhibit diminished overall life satisfaction and attenuated satisfaction levels concerning sleep quality and health status. Furthermore, men in such couples report reduced satisfaction with family, while women report lower satisfaction with both market work and housework. Subsequently, the authors split the sample into West and East German cohorts to analyze the impact of disparate social norms on the outcomes above. They find that couples in West Germany mainly drive the results.

The study employs established empirical methodologies and existing data to explore the influence of relative spousal income on life satisfaction. Its primary contribution lies in comparing West and East Germany, which are presumed to adhere to differing social norms. This topic is relevant, and the analysis seems appropriately conducted. However, I have some reservations regarding the drawn conclusions and the presentation of results, which I will detail further.

Comments:

1. The primary contribution of this study, as indicated by the manuscript’s title, lies in its examination of how gender norms influence the relationship between within-couple earner roles and well-being. However, this crucial finding is relegated to a supplementary position in the results section, with the associated table relegated to the Appendix. In contrast, the authors allocate most of their attention to presenting results on various well-being domains, yielding mixed and somewhat convoluted findings. I recommend that the authors reorganize their paper around their principal contribution to enhance clarity and coherence.

2. I suggest that the authors streamline the number of well-being metrics to focus on the most significant ones. Alternatively, they could construct an index measure of well-being (although, I guess, overall life satisfaction serves as a summary statistic for the other subdomains). This would allow the authors to focus on their main contribution (see my previous point).

3. Regarding the conclusion drawn, I hold reservations as I do not entirely agree with the authors’ conclusion that observations from West Germany predominantly drive all results. Notably, there appears to be no significant East-West disparity in overall life satisfaction for females, which I perceive as the most crucial summary statistic for well-being. I am intrigued by this observation and seek further clarification on why social norm disparities influence outcomes for men but not women. Clarification on this matter would enrich the understanding of the study’s findings.

4. The authors frequently state they investigate “how relative spousal income affects the well-being... of couples.” I find this statement misleading because, if I understand correctly, the authors only observe one partner and their reported satisfaction levels rather than both partners simultaneously. Consequently, the study provides insights into an individual’s well-being within couples. However, it does not directly address the well-being of “the couple” as an aggregated measure of both partners’ well-being.

6. PLOS authors have the option to publish the peer review history of their article (what does this mean?). If published, this will include your full peer review and any attached files.

Reviewer #1: No

Reviewer #2: No

---

## [Author Response · Author response to Decision Letter 0]

12 May 2024

Please find our response in the attached pdf or below

Reply to Comments by the Reviewers:

We are very grateful to Reviewer #1 and Reviewer #2 for many insightful comments and suggestions, that have helped us improve the quality of the paper. We provide detailed replies below. 

Reviewer #1:

1. “The regression model reported in equation (1) only includes the dummy for being above the cut-off and the running variable. It does not include a constant term and an interaction between the dummy and the running variable. I think both are (and should be) typically included in a Regression-Discontinuity design unless specific assumptions are made (see for example Imbens and Lemieux 2008 or Calonico et al. 2017 - which the authors also refer to). It’s thus not clear whether the estimates reported in the paper are derived from equation (1), which I don’t think identifies the desired effect, or from using the “rdrobust” package from Calonico and co-authors, which would include the two terms and therefore there is a mistake in the reported estimated equation.”

Authors: We thank the Reviewer for pointing this out. We apologize for the confusion. Throughout the entire analysis, we used the “rdrobust” package from Calonico and coauthors. Therefore, the Equation (1) was not complete. We have revised the Equation (1) accordingly adding the running variable and the interaction between the running variable and the dummy variable for individuals living in households in which the wife earns more than the husband. We do not report the constant term in Equation (1), since – following the suggestion of the Reviewer - we have now included the individual fixed effects in our main specification. 

2. “My second concern is related to the fact that people can endogenously sort on the two sides of the cut-off or that the estimates could reflect, for example, reverse causality.

a. The authors alleviate the concern by conducting a balancing check on both covariates (measured in the first year of marriage) and outcomes (as measured before marriage). However, it’s not entirely clear to me how this check is conducted. Given the panel structure of the data, the same person could be above or below the cut-off at different points in time. In which period is the running variable measured to decide whether a person is part of the ‘treated’ or ‘control’ group in the balancing check? Is the running variable also measured in the first year of marriage?”

Authors: We agree with the Reviewer that we were not clear on this point. Therefore, we have discussed it more in detail and we have now clarified that to conduct the balancing test we used the covariates measured in the first year of marriage, and the outcomes measured in the first year available in the SOEP before marriage. Following Getik (2022), the running variable is also measured in the first year of marriage. Furthermore, as suggested by the Reviewer, we have included in the Appendix an alternative balancing check that uses all the available years in the data for constructing the running variable (see Panel B of Table A.2, and column 2 of Table A.3). Reassuringly, we find that baseline covariates and baseline outcomes are balanced both when constructing the running variable using the first year of marriage or when considering all the available years in the data.

In the revised version of the paper, we clarify this point in the Section Empirical Specification (see pages 3 and 4).

“b. I think that the inclusion of individual fixed-effects also help alleviating this second concern, but it is currently left as a robustness exercise. I think it should be presented as the main analysis because (i) it accounts for the fact that unobserved heterogeneity might be correlated with sorting (ii) it exploits variation over time and within individual regarding the placement around the cut-off.”

Authors: We agree with this point. Following the suggestion of the Reviewer, in Equation (1) we now include individual fixed effects, which absorb the influence of any time-invariant individual heterogeneity. Therefore, all the estimates now control for individual fixed effects. 

In the text, we address this point in the Section Empirical Specification (see pages 3 and 4).

3. “Moreover, point estimates and significance levels often change considerably when including fixed-effects, to the point that the interpretation of the results changes. For example, I find it difficult to reconcile the large, negative and significant effect on life satisfaction for men with the non-significant effect on work satisfaction – if the story is really about relative earned income. When individual fixed-effects are included, instead, the estimate is more precise and there is evidence of a negative and significant effect on work satisfaction, which seems more in line with the proposed story. On the other hand, the estimated effect on mental health for men changes considerably and it is zero when individual fixed-effects are included. Similarly, after including fixed-effects, there is little evidence of a negative effect on housework, work and sleep satisfaction for women.”

Authors: We agree with the Reviewer. Therefore, the entire analysis now controls for individual fixed effects.

In the manuscript, we clarify this point in the Section Empirical Specification (see pages 3 and 4).

4. “The authors claim that “For both men and women, the effects on life satisfaction and mental well-being are mostly driven by respondents in West Germany”, but it seems that the results for men are not so conclusive.

a. The point estimates for life satisfaction for men in East and West Germany are virtually identical in Table A.1 (-0.079 vs. -0.082), even if only significant for West Germany (also, the East German sample is much smaller, which could explain why the estimate is less precise). It’s also surprising that the estimate for the entire sample (-0.139) it's much smaller than the ones for East and West Germany and does not lie between them. I cannot find the estimates by region when individual fixed-effects are included.

b. For mental health, the point estimate in Table A.2 for men it’s even larger in magnitude in East Germany (-0.061 vs. -0.037), and not significant in both cases.”

Authors: We thank the Referee for highlighting this point. Following the comments of both Reviewers, we have revised this part of the paper and substantially softened the language when discussing the heterogeneity of the effect of interest between East and West Germany.

The results of this analysis are now reported in Table 2. When considering the effect on life satisfaction for men, we find no evidence of significant differences between East and West Germany (see column 1 of Panel A). However, when considering women, we find that the negative effects on life satisfaction are significantly more pronounced in West Germany (see column 1 of Panel B). Moreover, while for men the effects on satisfaction with work, satisfaction with health as well as mental and physical health are not significantly different between East and West Germany (see columns 2 to 5 of Panel A), for women residing in West Germany we find significant negative effects on satisfaction with work and satisfaction with health and no evidence of significant differences in mental and physical health (see columns 2 to 5 of Panel B). We view these results as partially consistent with previous studies documenting substantial cultural differences in gender norms and female labor supply between East and West Germany and with previous work analyzing how gender identity affects female labor supply among women in West Germany. 

In the revised version of the paper, we now clarify these points in the updated Results Section (see pages 4 and 5).

5. “In Section 2 the authors write “We report the estimates obtained using the triangular kernel to recover the average effects using the full bandwidth ….” However, in the notes to the tables, they write “For each outcome, we used the optimal bandwidth using the mean square error (MSE) selection criteria”. It’s thus not clear which bandwidth is used, or why we would want to use a non-optimal one.”

Authors: We apologize for the confusion. Throughout the entire analysis, we are using the optimal bandwith as obtained from the “rdrobust” package from Calonico and coauthors. 

In the revised text, we now clarify this point in the Section Empirical Specification (see pages 3 and 4). 

Reviewer #2:

1. “The primary contribution of this study, as indicated by the manuscript’s title, lies in its examination of how gender norms influence the relationship between within-couple earner roles and well-being. However, this crucial finding is relegated to a supplementary position in the results section, with the associated table relegated to the Appendix. In contrast, the authors allocate most of their attention to presenting results on various well-being domains, yielding mixed and somewhat convoluted findings. I recommend that the authors reorganize their paper around their principal contribution to enhance clarity and coherence.” 

Authors: We agree with the Reviewer and have now revised the text to streamline our contribution. Following the suggestion of the Reviewer, we have decided to discuss more prominently the results on the heterogeneity of the effects between East and West Germany and have now reported the table in the main analysis (see Table 2). 

Furthermore, to improve clarity, we have narrowed down the number of outcomes presented and discussed in the main text, focusing on satisfaction with life, satisfaction with work, satisfaction with health, mental health, and physical health (see Table 1). The additional set of outcomes related to satisfaction with family, satisfaction with housework, satisfaction with sleep, satisfaction with childcare, and poor health are now presented in the Appendix (see Tables A.4 and A.6). Similarly for the figures, we have now reduced the number of outcomes presented in the main text focusing on satisfaction with life and satisfaction with health (see Figures 2 and 3), but have reported the figures for the other outcomes in the Appendix (see Figures A.1 and A.2).

2. “I suggest that the authors streamline the number of well-being metrics to focus on the most significant ones. Alternatively, they could construct an index measure of well-being (although, I guess, overall life satisfaction serves as a summary statistic for the other subdomains). This would allow the authors to focus on their main contribution (see my previous point).”

Authors: Thanks for the comment. In accordance with the Reviewer’s suggestion, we have reduced the number of outcomes presented and discussed in the main text. We have kept the secondary outcome variables in the Appendix (see also our response to point 1).

3. “Regarding the conclusion drawn, I hold reservations as I do not entirely agree with the authors’ conclusion that observations from West Germany predominantly drive all results. Notably, there appears to be no significant East-West disparity in overall life satisfaction for females, which I perceive as the most crucial summary statistic for well-being. I am intrigued by this observation and seek further clarification on why social norm disparities influence outcomes for men but not women. Clarification on this matter would enrich the understanding of the study’s findings.”

Authors: We thank the Referee for raising this point. Following the comments of both Reviewers, we have revised this part of the paper and substantially softened the language when discussing the heterogeneity of the effect of interest between East and West Germany.

The results of this analysis are now reported in Table 2. When considering the effect on life satisfaction for men, we find no evidence of significant differences between East and West Germany (see column 1 of Panel A). However, when considering women, we find that the negative effects on life satisfaction are significantly more pronounced in West Germany (see column 1 of Panel B). Moreover, while for men the effects on satisfaction with work, satisfaction with health as well as mental and physical health are not significantly different between East and West Germany (see columns 2 to 5 of Panel A), for women residing in West Germany we find significant negative effects on satisfaction with work and satisfaction with health and no evidence of significant differences in mental and physical health (see columns 2 to 5 of Panel B). We view these results as partially consistent with previous studies documenting substantial cultural differences in gender norms and female labor supply between East and West Germany and with previous work analyzing how gender identity affects female labor supply among women in West Germany. 

In the revised version of the paper, we now clarify these points in the updated Results Section (see pages 4 and 5).

4. “The authors frequently state they investigate “how relative spousal income affects the well-being... of couples.” I find this statement misleading because, if I understand correctly, the authors only observe one partner and their reported satisfaction levels rather than both partners simultaneously. Consequently, the study provides insights into an individual’s well-being within couples. However, it does not directly address the well-being of “the couple” as an aggregated measure of both partners’ well-being.”

Authors: We thank the Reviewer for this comment. We agree that the text was indeed misleading. Therefore, we have revised it accordingly.

---

## [Decision Letter · Decision Letter 1]

12 Jun 2024

Relative Income within the Household, Gender Norms, and Well-being

PONE-D-24-02294R1

Dear Dr. Giuntella,

We’re pleased to inform you that your manuscript has been judged scientifically suitable for publication and will be formally accepted for publication once it meets all outstanding technical requirements.

Kind regards,

José Alberto Molina

Academic Editor

PLOS ONE

Additional Editor Comments (optional):

Reviewers' comments:

Reviewer's Responses to Questions

**Comments to the Author**

1. If the authors have adequately addressed your comments raised in a previous round of review and you feel that this manuscript is now acceptable for publication, you may indicate that here to bypass the “Comments to the Author” section, enter your conflict of interest statement in the “Confidential to Editor” section, and submit your "Accept" recommendation.

Reviewer #1: All comments have been addressed

Reviewer #2: All comments have been addressed

2. Is the manuscript technically sound, and do the data support the conclusions?

Reviewer #1: Yes

Reviewer #2: (No Response)

3. Has the statistical analysis been performed appropriately and rigorously? 

Reviewer #1: Yes

Reviewer #2: (No Response)

4. Have the authors made all data underlying the findings in their manuscript fully available?

Reviewer #1: Yes

Reviewer #2: (No Response)

5. Is the manuscript presented in an intelligible fashion and written in standard English?

Reviewer #1: Yes

Reviewer #2: (No Response)

6. Review Comments to the Author

Reviewer #1: (No Response)

Reviewer #2: (No Response)

7. PLOS authors have the option to publish the peer review history of their article (what does this mean?). If published, this will include your full peer review and any attached files.

Reviewer #1: No

Reviewer #2: No

---

## [Editor Report · Acceptance letter]

26 Aug 2024

PONE-D-24-02294R1 

PLOS ONE

Dear Dr. Giuntella, 

I'm pleased to inform you that your manuscript has been deemed suitable for publication in PLOS ONE. Congratulations! Your manuscript is now being handed over to our production team.

Kind regards, 

on behalf of

Professor José Alberto Molina 

Academic Editor

PLOS ONE